# Fully Dynamic Consistent Facility Location

**Vincent Cohen-Addad,**
CNRS & Sorbonne Université
vcohen@di.ens.fr

**Niklas Hjuler,**
University of Copenhagen
hjuler@di.ku.dk

**Nikos Parotsidis,**
University of Copenhagen
nipa@di.ku.dk

**David Saulpic,**
Ecole normale supérieure
Sorbonne Univeristé
david.saulpic@lip6.fr

**Chris Schwiegelshohn**
Sapienza University of Rome
schwiegelshohn@diag.uniroma1.it

## Abstract

We consider classic clustering problems in fully dynamic data streams, where data elements can be both inserted and deleted. In this context, there are several important parameters: (1) the quality of the solution after each insertion or deletion, (2) the time it takes to update the solution, and (3) how different consecutive solutions are. The question of obtaining efficient algorithms in this context for facility location, $k$-median and $k$-means has been raised in a recent paper by Hubert-Chan et al. [WWW'18] and also appears as a natural follow-up on the online model with recourse studied by Lattanzi and Vassilvitskii [ICML'17] (i.e.: in insertion-only streams).

In this paper, we focus on general metric spaces and mainly on the facility location problem. We give an arguably simple algorithm that maintains a constant factor approximation, with $O(n \log n)$ update time, and total recourse $O(n)$. This improves over the naive algorithm which consists in recomputing a solution after each update and that can take up to $O(n^2)$ update time, and $O(n^2)$ total recourse. Our bounds are nearly optimal: in general metric space, inserting a point takes $O(n)$ times to describe the distances to other points, and we give a simple lower bound of $O(n)$ for the recourse. Moreover, we generalize this result for the $k$-medians and $k$-means problems: our algorithms maintain a constant factor approximation in $\widetilde{O}(n + k^2)$ time per update.

We complement our analysis with experiments showing that the cost of the solution maintained by our algorithm at any time $t$ is very close to the cost of a solution obtained by quickly recomputing a solution from scratch at time $t$ while having a much better running time.

## 1 Introduction

Clustering is a core procedure in unsupervised machine learning and data analysis. Due to the large number of applications, clustering problems have been extensively studied for several decades. The existing literature includes both very precise algorithms[1, 18, 31], and very fast ones [34]. Due to the importance of the task, clustering problems have also been studied in several computing settings, such as the streaming model [11] and the sliding-window model [7], the distributed model [4], in the dynamic model [24], and others.

Applications nowadays operate on dynamically evolving data, e.g., pictures are constantly added and deleted from picture repositories, purchases are continuously added into online shopping systems, reviews are added or being edited in retail systems, etc. Due to the scale and the dynamic nature

of the data at hand, conventional algorithms designed to operate on static inputs become unable to handle the task for two main reasons. First, the running time of even the most efficient algorithms is too expensive to execute after every single change in the input data. Second, re-running a static algorithm after every update might generate solutions that differ substantially between consecutive updates, which might be undesirable for the application at hand. The number of changes in the maintained solution between consecutive updates is called the *recourse* of the algorithm. Our study is motivated by these limitations of static algorithms, or dynamic algorithms that are effective on only one of the two objectives.

Most fundamental problems in computer science have been studied in the dynamic setting. In a very high-level, a dynamic algorithm computes a solution on the initial input data and as the input undergoes insertions or/and deletions of elements, the algorithm updates the solution to reflect the current state of the data. A dynamic algorithm may allow only insertions or only deletions, or may support an intermixed sequence of insertions and deletions, in which case the algorithm is called fully dynamic. The running time of a dynamic algorithm can either guarantee a worst-case update time after each update, or a bound on the average update time over a sequence of updates which is called amortized update bound. A dynamic algorithm with worst-case update bounds is the most desirable, and often hard to obtain, but in several applications algorithms with amortized update bounds are sufficient.

Most of the clustering-related literature has focused on the online model where the updates are restricted to insertions only and a decision cannot be revoked or on the streaming model where there is a specific memory budget not to be exceeded. However, as observed by Lattanzi and Vassilvitskii [30] the online model may appear too restrictive: if a bad decision has been made, it is often fine to spend some time to correct it instead of suffering the bad decision (i.e.: keeping a bad clustering) for the rest of the stream. However, spending too much time on the modification of the clustering may be counterproductive and that's what we aim at capturing in the fully-dynamic model with limited recourse: keeping a good clustering by spending the least among of time and making as few changes to the current clustering as possible.

In this paper, we study fully-dynamic algorithms for classic clustering problems. In particular, we consider the facility location, the $k$-means, and the $k$-median problems in the dynamic setting. In the static case, these problems are defined as follows. Let $X$ be a set of $n$ points, and $d : X \times X \to \mathbb{R}$ a distance function. We assume that $d$ is symmetric and that $(X, d)$ forms a metric space. For the $(k, p)$-clustering problem, the objective function that we seek to optimize is $C_p(X, S)$, where $S \subseteq X, |S| = k$. Setting $p = 1$ gives the $k$-median objective, and $p = 2$ the $k$-means one. For the facility location problem the objective function is $C(X, S)$.

$$C(X, S) := \sum_{x \in X} \min_{c \in S} d(x, c) + f \cdot |S| \qquad C_p(X, S) := \sum_{x \in X} \min_{c \in S} d^p(x, c),$$

All of these problems are NP-Hard, so our best hope is to design algorithms with provable approximation guarantees. There is an extensive line of work on algorithms with constant approximation guarantees for all three aforementioned problems [1, 3, 10, 20, 26, 27, 31, 33]. On the other hand, Mettu and Plaxton [34] showed an $\Omega(nk)$ lower bound for computing an $O(1)$-approximate solution for k-median and k-means in general metric spaces.

In the dynamic setting, the goal is to maintain efficiently a good solution to the clustering problem at hand as the set of points undergoes element insertions and deletions. The main criterion for designing a *good* dynamic algorithm for these problems is the quality of the clustering, with respect to the optimum solution, at any given time. However, in many applications, it is equally important to maintain a *consistent* clustering, namely a clustering with bounded recourse. Lattanzi and Vassilvitskii [30] have recently considered consistent clustering problems in the online setting, where the points appear in sequence and the objective is to maintain a constant factor approximate solution while minimizing the total number of times the maintained solution changes over the whole sequence of points. Another criterion that has been explored much less but which is highly important when dealing with massive data is the time it takes to update the solution after each update so that the solution remains within a constant factor from the optimum solution.

## 1.1 Our Contribution

We present the first work that studies fully-dynamic algorithms while considering the approximation guarantee, consistency and update time, all at the same time. From an input perspective, we consider general metric spaces. Thus, an element of the input is a point in a metric space which is defined by the distances to the other points of the metric. The contribution of our paper is summarized as follows:

• We give a fully-dynamic algorithm for the facility location problem that maintains a constant factor approximate solution with constant recourse per update and $O(n \log n)$ update time. We moreover show that constant recourse per update is necessary for achieving a constant factor approximation.

• We extend the algorithm for facility location to the $k$-median and $k$-means problems. Here, our algorithm maintains a constant factor approximate solution with $\widetilde{O}(n + k^2)$[1] update time (Theorem 3.1). This is the first non-trivial result for these problems, as the only known solution for these problems was to recompute from scratch after each update: this requires time $\Omega(nk)$ for $k$-median and $\Omega(n^2)$ for facility location, per update. Hence, our time bounds are significantly better than the naive approach for a large range of $k$.

**Empirical Analysis.** We complement our study with an experimental analysis of our algorithm on three real-world data sets and show that it outperforms the standard approach that recomputes a solution from scratch after each update using a fast static algorithm. Interestingly, we show that this barely impacts the approximation guarantee. At the same time, our algorithm outperforms by at least three orders of magnitude the simple-minded solutions, both in terms of running time and total number of changes made in the maintained solution throughout the update sequence.

## 1.2 Related Work

**Online and Consistent Clustering.** Online algorithms for facility location were first proposed by Meyerson [35] in his seminal paper. Fotakis [16] later showed that the algorithm has a competitive ratio of $O(\log n / \log \log n)$, which is also optimal. Additionally, the algorithm has a constant competitive ratio if the points arrive in a random order [35, 29]. There also exist $O(\log n)$ competitive deterministic algorithms, see [2, 15]. This was recently extended to the online model that incorporates deletions [12].

Online algorithms for clustering that are only allowed to place centers cannot be competitive. This led to the consideration of the incremental model, which allows two clusters to be merged at any given time. Work in this area includes [9, 14]. The number of reassignments (commonly referred to as *recourse*) over the execution of an incremental algorithm may be arbitrary. However, recently, Lattanzi and Vassilvitskii [30] considered the online clustering problem with bounded total recourse. They showed a lower bound of $\Omega(k \log n)$ changes over an arbitrary sequence of updates, and presented an algorithm that can maintain a constant factor approximation while limiting the total recourse to $O(k^2 \cdot \log^4 n)$. Their work differs to ours in that elements can only be added, and that they do not consider optimizing the running time. In the fully dynamic case their bound on the recourse does not hold, and we moreover show that constant recourse per update is unavoidable.

**Fully-Dynamic and Streaming Algorithms.** Streaming algorithms for clustering can be used to obtain fast dynamic algorithms by recomputing a clustering after each update. Since streaming algorithms are highly memory compressed and typically process updates in time linear in the memory requirement, the approach automatically yields good update times. Low-memory adaptations of Meyerson's algorithm [35] turned out to be simple and particularly popular, see [8, 29, 37]. Another technique for designing clustering algorithms in the streaming models is by maintaining coresets, see the following recent survey for an overview [36]. For fully dynamic data streams, the only known algorithms for maintaining coresets for $k$-means and $k$-median in Euclidean spaces using small space and update times are due to Braverman et al. [6] and Frahling and Sohler [17]. There also exists some work on estimating the cost of Euclidean facility location in dynamic data streams, see [13, 25, 28].

For more general metrics, the problem of maintaining a clustering dynamically has been considered by Henzinger et al. [22] and Goranci et al. [19] who consider the facility location in bounded doubling

dimension. The arguably most similar previous work to ours is due to Hubert-Chan et al. [24]. They consider the $k$-center problem in general metrics in the fully dynamic model. Here, they were able to maintain a constant factor approximation with update time $O(k \log n)$ [2]. Whether an algorithm in the fully dynamic model with low recourse and update times exists, was left as an open problem.

## 1.3 Preliminaries

We assume that we are given some finite metric space $(X, d)$, where $X$ is the set of points and $d : X \times X \to \mathbb{R}_{\geq 0}$ a distance function. Every entry $d(a, b)$ is stored in a (symmetric) $n \times n$ matrix $D$. Our algorithms work in the distance oracle model, which assumes that we can access any entry of $D$ in constant time.

Our input consists of tuples $(X, \mathbb{R}_{\geq 0}^n, \{-1, 1\})$. The first coordinate is the identifier of some point $p \in X$, the second coordinate is the column/row vector in $D$ associated with $p$, and the last coordinate signifies insertion $(1)$ or deletion $(-1)$. We assume that the stream is consistent, which means that no point can be deleted without having been previously inserted. The adversary generating the point sequence is called adaptive if he can modify the sequence depending on the algorithm's choices. Throughout the paper, we let $X^t$ be the set of points present at time $t$, $n$ be the total number of updates, and $n^* := \sup_{t \in 1, \ldots, n} |X^t|$ be the maximum number of points present at the same time. We denote by $OPT^t$ the optimum solution at time $t$. All our results could be phrased in term of $|X^t|$, but for simplicity we present them in terms of $n^*$.

**Roadmap.** Our paper is organized as follows. In Section 2, we describe our algorithm for fully dynamic facility location. Section 3 extends these results to $k$-median and $k$-means clustering. We conclude with an experimental evaluation of our algorithms in Section 4 on real-world benchmarks. All omitted proofs can be found in the supplementary material.

## 2 Dynamic Facility Location

The goal of this section is to prove the following theorem.

**Theorem 2.1.** *There exist a randomized algorithm that, given a metric space undergoing insertion and deletions of points, maintains a set of center $S^t$ such that :*

- *each update is processed in time $O(n^* \log(n^*))$ with probability $1 - 1/n^*$*

- *at any given time $t$, $C(X^t, S^t) = O(1) \cdot C(X^t, OPT^t)$ with probability $1 - 1/n^*$*

- *$\sum_{t=0}^{n} |S^t \triangle S^{t+1}| = O(n)$, i.e. the amortized recourse is $O(1)$ per update.*

The proof is divided into several lemmas: we first study how the optimal cost behaves upon dynamic updates, and we exhibit then an algorithm that maintains a solution whose cost evolves in a similar way as the optimum.

Although, perhaps counter intuitive, removing a point from the input in a finite metric may increase the cost of a clustering, if one cannot locate a center there anymore. We show in the supplementary material that this increase is bounded by a factor 2: this leads to the following lemma, which bounds the evolution of the optimal cost.

**Lemma 2.2.** *Let $OPT_{before}$ be the optimal cost of an initial metric space $X$. After an arbitrary sequence of $n_i$ insertions and $n_d$ deletions of points in $X$, the optimal solution $OPT_{after}$ satisfies $OPT_{before}/2 - n_d \cdot f \leq OPT_{after} \leq 2(OPT_{before} + n_i \cdot f)$*

**Maintaining a solution during a few updates.** We now turn on designing an algorithm competitive with the optimal solution, showing first how to deal with a small number of updates. In order to process deletions, we define the notion of *substitute* centers: given a function $s$ mapping every center from the initial solution to a center in the current one, we say that $s(c)$ substitutes $c$. Initially, $s(c) = c$. When a center $c$ is deleted, the algorithm opens a replacement center $c_r$, and updates the function $s$: $s(s^{-1}(c)) = c_r$.

The algorithm is as follows: when a point $x$ is inserted, we open as a facility at $x$, and for convenience we define $s(x) = x$. When a point $x$ is deleted, we have two cases: either $x$ was not an open facility, in which case the algorithm does nothing, or $x$ was a facility. In the latter case, let $c = s^{-1}(x)$ : the algorithm opens the closest point $c'$ of $c$ in $X^0$ that is still part of the metric, and set $s(c) = c'$. This choice of $c'$ ensures that, for all points $x$ in the current metric space, $d(c', c) \leq d(x, c)$.

**Lemma 2.3.** *Starting from any metric space $(X^0, d)$ and an $\alpha$-approximation with cost $\Theta$, the algorithm described above maintains a $(8\alpha + 4)$-approximation during $\frac{\Theta}{4\alpha f}$ updates, with $O(1)$ recourse and $O(n^*)$ time amortized per update.*

*Proof.* This algorithm opens at most one new facility at every update: the recourse is thus at most 1. The time to process an insertion is constant, and the time to process a deletion is at most $O(n^*)$ (the time required to compute the closest point to $x$).

We now analyse the cost of the solution produced after $t$ updates. Since the recourse is at most 1 per update, the cost of open facilities increases by at most $t \cdot f$. Since every inserted points is opened as a center, it does not contribute to the connection cost: this cost changes therefore only by deletions of points from $X^0$. Similarly to Lemma 2.2, one can show that the connection cost of a point $x \in X^0$ at most doubles. More formally, let $c \in X^0$ be the center that serves $x$ in the initial solution. Let $c' = s(c)$ be the center that substitutes $c$ in the current metric. By the choice of $c'$ and triangle inequality, it holds that $d(x, c') \leq d(x, c) + d(c, c') \leq 2d(x, c)$. Hence, the total serving cost is at most twice as expensive as in the initial solution. The cost at time $t$ is therefore at most $2\Theta + t \cdot f$.

Let $t \leq \frac{\Theta}{4\alpha f}$, and $\text{OPT}^0$ the optimal cost in the initial state. By Lemma 2.2, the optimal cost at time $t$ is at least $\text{OPT}^0/2 - t \cdot f \geq \text{OPT}^0/4$, since by definition of $\alpha$ it holds $t \leq \frac{\text{OPT}^0}{4f}$. Moreover, the cost of our algorithm at time $t$ is at most $\Theta(2 + \frac{1}{\alpha}) \leq \text{OPT}^0(2\alpha + 1)$. Combining the two inequalities gives that our algorithm is a $(8\alpha + 4)$-approximation for all $t \leq \frac{\Theta}{4\alpha f}$, which concludes the proof. $\square$

We remark that the parameters can be optimized: for instance, with a suitable data structure, the time to find a substitute center can be logarithmic; however, this is dominated by the complexity of finding the initial $\alpha$-approximation.

**Maintaining a solution for any number of updates.** We combine Lemma 2.3 with a classic static $O(1)$-approximation algorithm, namely Meyerson's algorithm, to prove Theorem 2.1.

*Proof of Theorem 2.1.* We summarize here the useful properties of Meyerson algorithm, and refer to the supplementary material for more details. The algorithm processes the input points in a random order, opening each point $x$ with probability $d(x, F)/f$ (where $F$ is the set of previously opened facilities). If the algorithm opens $k$ facilities, its running time is $O(kn^*)$, and the cost is at least $\Theta \geq kf$. Hence, the running time is $O(\Theta/f \cdot n^*)$. Moreover, one can assume that the cost is always at most $n^* f$ (by opening a facility at every point).

We say that a run of Meyerson's algorithm is *good* if it yields a $O(1)$-approximate solution. By the analysis in [35], a run is good in expectation (where the randomness comes from the random ordering of points): hence, by running $\log(2n^*)$ independent copies of the algorithm, at least one run is good with probability $1 - (1/n^*)^2$. We let $\alpha$ be the approximation constant of this algorithm.

Therefore, our main algorithm works as follows: start with a solution given by Meyerson's algorithm of cost $\Theta$, use Lemma 2.3 to maintain a solution during $\frac{\Theta}{4\alpha f}$ updates, and then recompute from scratch. We call the intervals between consecutive recomputations *periods*, and note that they are random objects: the length of a period is determined by the cost of its initial solution, which is a random variable.

We first analyze the running time of this algorithm. Within one period, Lemma 2.3 ensures that the running time is $O(n^*)$ per update. Moreover, the running time of the initial recomputation is $O(\Theta/f \cdot n^* \log n^*)$, and the length of the period is $\Omega(\Theta/f)$. Therefore the amortized running time is $\widetilde{O}(n^*)$ per update. Since the initial recourse is $O(\Theta/f)$, the same argument proves that the recourse is amortized $O(1)$ per update.

We aim at using again Lemma 2.3 to prove that, at a given time $t$, the solution is a constant factor approximation. For this, let $P$ be the period in which $t$ appears. If the period is good, then Lemma 2.3

concludes. Unfortunately, the fact that $t$ is in $P$ is not independent of $P$ being good (for instance, if $P$ is very long it is unlikely to be good). However, note that the starting time of $P$ cannot be before $t - n^*$: indeed, a period lasts for at most $\frac{\Theta}{4\alpha f} \le \frac{n^* f}{4\alpha f} \le n^*$ updates. Hence, if we condition on all periods starting between $t - n^*$ and $t$, Lemma 2.3 applies and the solution at time $t$ is a constant factor approximation. Since any period is good with probability $1 - (1/n^*)^2$, all periods between $t - n^*$ and $t$ are good with probability $1 - 1/n^*$ by union bound. This concludes the proof. $\qquad\square$

The algorithm sketched in the previous proof can be transformed so that the complexity becomes $\widetilde{O}(n^*)$ in the worst case, by spreading the recomputation over several updates (see supplementary material). Moreover, randomization is not needed in order to maintain the solution (only to compute a starting approximation): hence the algorithm works against an adaptive adversary.

We conclude this section by showing that our algorithm is (up to a logarithmic factors) optimal both for update time and recourse.

**Proposition 2.4.** *Any algorithm maintaining a $O(1)$-approximation for Facility Location must have an amortized update time $\Omega(n^*)$ and total recourse $\Omega(n)$, where $n$ the total number of updates.*

# 3   Dynamic $k$-Median and $k$-Means in Linear Time

In this section we adapt the algorithm from Section 2 to handle the stricter problems of $k$-Median and $k$-Means. For simplicity, we call $(k, p)$-clustering the problem of finding $k$ centers that minimize $C_p$ ($p = 1$ for $k$-Median and $p = 2$ for $k$-Means).

Roughly speaking, our algorithm works as follows. We use an adaptation of the algorithm from Section 2 to maintain a coreset $\mathcal{R}$ of $\widetilde{O}(k)$ points that contain a constant factor approximate solution for the $(k, p)$-clustering problem. Then, we apply a constant factor approximation algorithm for the metric $(k, p)$-clustering problem on the maintained coreset (e.g., we can use a quadratic-time local-search algorithm, see [21]). This yields the following theorem.

**Theorem 3.1.** *There exists a randomized algorithm that, given a metric space undergoing insertions and deletions of points, maintains a set of centers $S^t$ with $\widetilde{O}(n^* + k^2)$ update time such that, for any time $t$, $C_p(X^t, S^t) = O(1) \cdot C_p(X^t, OPT^t)$.*[3]

The remainder of this section is devoted in proving Theorem 3.1. The main hurdle in applying the framework from Section 2 is that the optimum solution can change drastically with the addition or deletion of a point, and it is therefore not easy to adapt the previous amortization argument. To overcome this barrier, we make use of the following lemma, from [9] and [30]:

**Lemma 3.2.** *Let $L$ be some integer. With probability $1/2$, running Meyerson's algorithm for Facility Location with $f = \frac{L}{k(1 + \log n^*)}$ gives a set $S$ of $4k \cdot (1 + \log n^*) \cdot (2^{2p+1} \cdot \frac{C_p(X, OPT)}{L} + 1)$ centers such that $C_p(X, S) \le L + 4 \cdot C_p(X, OPT)$.*

For completeness, we provide the pseudocode of Meyerson's algorithm, adapted for our purpose, in Procedure *MeyersonCapped*. The lemma implies that, if we know a value $L$ that approximates OPT within a factor 2, Procedure *MeyersonCapped* computes a set of points $\mathcal{R}$ and an assignment of points $\phi$ such that $\sum_{x \in X} d(x, \phi(x)) \le 6 \cdot C_p(X, \text{OPT})$ with probability $1/2$. This probability can be boosted to $1 - (1/n^*)^2$ by taking the union of $q = O(\log n^*)$ independent copies of the algorithm. Therefore for all $i = 1, ..., q$, our algorithm will use this lemma assuming $C_p(X, \text{OPT}) \in [2^i, 2^{i+1})$, and taking $L = 2^i$. This provides, for all $i$, a set $\mathcal{R}_i$ of $O(k \log^2 n^*)$ centers.

It remains to maintain those sets dynamically. Similarly to Section 2, we use the solution computed by Procedure *MeyersonCapped* for the subsequent $k$ updates, so that we can amortize the update-time bound. However, for $(k, p)$-clustering it is not possible to bound the cost of OPT after a few updates. We overcome this obstacle by updating the sets $\mathcal{R}_i$ more carefully. More precisely, let $\mathcal{R}_i^t$ be the (updated) set $\mathcal{R}_i$ after $t$ updates of the algorithm. The algorithm ensures the following invariant:

**Invariant 3.3.** *The set $\mathcal{R}_i^t$ has size $O(k \log^2 n^*)$ and, with high probability, there exists $i$ such that $C_p(X^t, \mathcal{R}_i^t) = O(1) \cdot C_p(X^t, OPT^t)$.*

**Input:** An integer $L$, a set of points $X$
**Output:** A set of centers $\mathcal{R}$, an assignment $\phi$ of point to centers, and $t_l$ the id of the last center opened
1: Let $\mathcal{R} \leftarrow \emptyset$ and $x_1, ..., x_{|X|}$ be a random order on the points of $X$
2: **for all** $i \in \{1, ..., |X|\}$ **do**
3:    **if** $|\mathcal{R}| < 4k \cdot (1 + \log n^*) \cdot (2^{2p+2} + 1)$ **then**
4:       add $x_i$ to $\mathcal{R}$ with probability $\frac{d(x_i, \mathcal{R})^p k (1 + \log n)}{L}$
5:       **if** $|\mathcal{R}| = 4k \cdot (1 + \log n^*) \cdot (2^{2p+2} + 1)$ **then**
6:          $t_l \leftarrow i$
7:       **end if**
8:    **end if**
9:    $\phi(x_i) \leftarrow \arg\min_{c \in \mathcal{R}} \{d(x_i, c)^p\}$
10: **end for**

(a) MeyersonCapped$(L, X)$

**Input:** Integers $L$ and $t_l$, a set of points $X$ and a set of centers $\mathcal{R}$
**Output:** Updated $\mathcal{R}$ and $t_l$, an assignment $\phi$ of points to centers
1: $\{t_l$ is the last time *MeyersonCapped* was invoked.$\}$
2: **for all** $j \in \{t_l, ..., |X|\}$ **do**
3:    **if** no center was opened yet **then**
4:       add $x_j$ to $\mathcal{R}$ with probability $\frac{d(x_j, \mathcal{R})^p k (1 + \log n^*)}{L}$
5:       **if** $x_j$ is added to $\mathcal{R}$ **then**
6:          $t_l \leftarrow j$, $\phi(x_j) \leftarrow x_j$
7:       **end if**
8:    **else** {Update $\phi$}
9:       $\phi(x_j) \leftarrow \arg\min_{z \in \{\phi(x_j), x_{t_l}\}} \{d(x_j, z)^p\}$
10:    **end if**
11: **end for**

(b) DeletePoint$(L, t_l, X, \mathcal{R})$. $L$ is the value with which we approximate $C_p(X, \text{OPT})$ and $t_l$ is the last time MeyersonCapped opened a center.

For this, initialize $\mathcal{R}_i$ to be the union of the outputs of $q = O(\log n^*)$ independent executions of *MeyersonCapped*$(L_i, X)$, for $L_i = 2^i$ and $i = 1, ..., q$. The algorithm updates these sets during $k$ updates before recomputing them from scratch. In the case of a point insertion, it suffices to add the new point to all $\mathcal{R}_i$: over $k$ updates, this changes the cardinality by at most $k$ while the cost remains the same, and therefore the two conditions are met. The case of a point deletion requires more work. The idea is, as in Section 2, to replace the deleted center by its closest point. However, this is not enough to ensure Invariant 3.3: this is taken care of by Procedure *DeletePoint*, which finds the next point in $X^t$ that *MeyersonCapped* would open, if there was no constraint on the size of $\mathcal{R}$.

We are now ready to describe our fully-dynamic algorithm for maintaining a constant-approximate solution to the $(k, p)$-clustering problem. The algorithm uses Procedures *MeyersonCapped* and *DeletePoint* as subroutines to build and maintain the sets $\mathcal{R}_i$ for $i$, and after each update calls the static constant-approximate algorithm to compute an approximate solution $S_i^t$ on each weighted instance $\mathcal{R}_i$ (where the weight of each point $x \in \mathcal{R}_i$ corresponds to the number of points of $X^t$ assigned to $x$ by the function $\phi_i$, computed in Procedures *MeyersonCapped* and *DeletePoint*. After each update, the algorithm keeps the solution $S_i^t$ that minimizes $C_p(\mathcal{R}_i^t, S_i^t)$, that is, $S^t = S_i^t$ for $i = \arg\min_i \{C_p(\mathcal{R}_i^t, S_i^t)\}$. The pseudocode of the algorithm is stated in Algorithm 1. The proof of Invariant 3.3 is stated in the supplementary material. The next lemma, together with Invariant 3.3, shows that $S^t$ can be used as a solution for the entire set $X^t$.

**Lemma 3.4.** *Let $OPT(\mathcal{R}_i^t)$ be the optimal solution in the weighted set $\mathcal{R}_i^t$. Then it holds that $C_p(X^t, OPT(\mathcal{R}_i^t)) \leq 2^{3p-1}(C_p(X, \mathcal{R}_i^t) + C_p(X^t, OPT^t))$.*

This proves the second part of Theorem 3.1: for $i$ such that $C_p(X^t, \mathcal{R}_i^t) = O(1) \cdot C_p(X^t, OPT^t)$, the solution computed on the set $\mathcal{R}_i^t$ is a good approximation of the optimal solution, and therefore the algorithm maintains a constant factor approximation. The bound on the running time being similar to the one of Section 2, we provide it in the supplementary material.

## 4 Empirical Analysis

In this section, we evaluate our algorithm for facility location experimentally. Recall that we aim to strike a balance between (1) overall running time, (2) the cost of the solution, (3) the total recourse. Our implementation follows the framework outline in Theorem 2.1. As part of the recomputation step between two periods, we run 5 independent executions of Meyerson's algorithm, and selecting the execution with lowest cost. The updates within a period are handled by assigning to closest center if distance is less than $f$ or otherwise open a new center at the point, and we simply remove a client if it gets deleted. We compare our algorithm against two variants of Meyerson's algorithm.

---

**Algorithm 1** Fully Dynamic $(k, p)$-clustering

---

**Input:** For all $t$, $X^t$ be the set of points at time $t$
**Output:** For all $t$, a set $S^t$ of centers at time $t$ and an mapping $\phi^t : X^t \to S^t$ of points to $S^t$
1: Let $\mathcal{R}_i^t$ be the coreset at time $t$ for $L_i$, and $\phi_i^t$ the function mapping every point of $X^t$ to its closest point in $\mathcal{R}_i^t$
2: Let $t_0$ be the last time $MeyersonCapped$ was called

3: **for all** time $t$ **do**
4:   **if** a point $x^t$ is inserted **then**
5:     Let $\mathcal{R}_i^t = \mathcal{R}_i^{t-1} \cup \{x^t\}, \forall i \in [\log n]$
6:   **else if** a point $x^t$ is removed **then**
7:     **for all** $i \leftarrow 1, ..., \log n$ **do**
8:       **if** $x^t \notin \mathcal{R}_i^{t-1}$ **then**
9:         $\mathcal{R}^t \leftarrow \mathcal{R}^{t-1}$
10:       **else**
11:         Let $z \leftarrow \phi^{t_0}(x^t)$
12:         Let $z' \in X^t$ be the closest to $z$ (breaking ties arbitrarily)
13:         $\mathcal{R}^t \leftarrow \mathcal{R}^{t-1} \cup \{z'\}$
14:         $\{t_l^i$ is the last time $MeyersonCapped$ was invoked on $\mathcal{R}^{t-1}\}$
15:         Call $DeletePoint(L_i, t_l^i, \mathcal{R}^{t-1})$
16:       **end if**
17:     **end for**
18:   **end if**
19:   **if** $t$ is a multiple of $k$ **then**
20:     {Recompute from scratch all $\mathcal{R}_i$}
21:     $t_0 \leftarrow t$
22:     **for all** $i \leftarrow 1, ..., \log n$ **do**
23:       $\mathcal{R}_i^t \leftarrow MeyersonCapped(L_i, X^t)$
24:     **end for**
25:   **end if**
26:   {Keep the best among all $\mathcal{R}_i^t$ and assignments $\phi^t$}
27:   $(i, S^t) \leftarrow \underset{i, C_i = A(\mathcal{R}_i^t)}{\arg\min} \; \mathbf{C}_p(C_i, \mathcal{R}_i^t)$
28:   Let $\psi : \mathcal{R}_i^t \to C_i$ be the assignment computed by $A$
29:   $\phi^t \leftarrow \psi \circ \phi_i^t$
30: **end for**

---

The first one, termed *MeyersonRec*, re-runs Meyerson at every single update. The second, termed *MeyersonSingle*, consists of a single execution of Meyerson for all updates, where deletions are handled by just removing the distance cost of the deleted point. Following Hubert-Chan et al. [24], we incorporate deletions by considering a sliding window over the data set. A point is inserted/deleted when it enters/exists the window, respectively.

**Data Set and Setup.** We consider the following data sets, equipped with the Euclidean distance.

• The Twitter data set [23], considered by [24], consists of 4.5 million geotagged tweets in 3 dimensions (longitude, latitude, timestamp). We restricted our experiments to the first 200K tweets.

• The COVERTYPE data set [5], considered by [30], from the UCI repository with 581K points and 54 dimensions. We restricted our experiments to the first 100K points and 10 dimensions (the ones we believed to be appropriate for an Euclidan metric).

• The USCensus1990 data set [32] from the UCI repository has 69 dimensions and 2.5 million points. We restricted our experiments to the first 30K points.

We restricted the number of points considered due to time constraints. Since larger data sets typically have more complicated ground truths, we used a larger windows for them containing more samples. To avoid overfitting, we also adjusted the cost of opening a facility depending on the window size, i.e. for larger windows a lower opening cost per facility. For COVERTYPE and USCensus1990, we used a window size of 5000 points and a facility cost of $0.5$; for Twitter, the window size was 10000 and the facility cost $0.004$. All our codes are written in Python. The experiments were executed on a Windows 10 machine with processor: Intel(R) Core(TM) i7-8700K CPU @ 3.70GHz, 3701 Mhz, 6 cores, 12 Logical processors, and 16 GB RAM.

**Results.** In all three data sets we generally observed the same behavior in terms of running time, cost, and the number of clusters opened, see Figure 2. Our algorithm is 100 times faster than MeyersonRec. Compared to MeyersonSingle, our algorithm is slower initially. When the number of processed points becomes very large, the running time of MeyersonSingle deteriorates comparatively, as it never removes a facility once it has been opened: the time to compute the distance to the

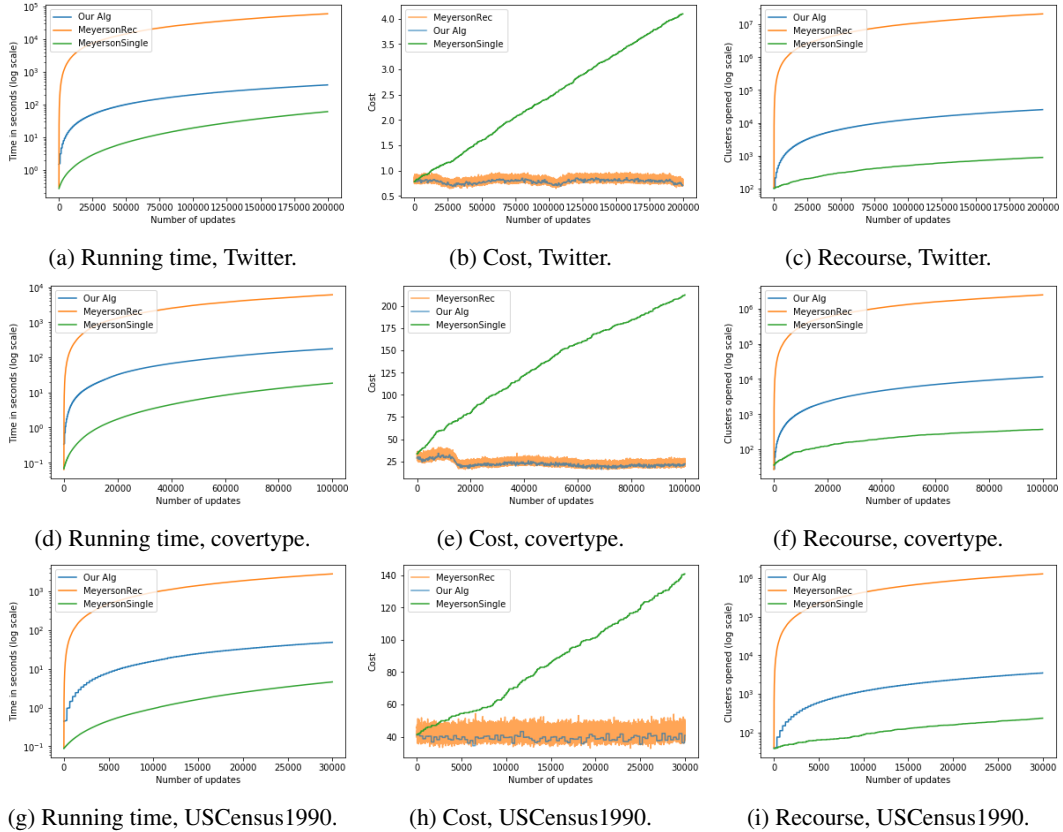

| (a) Running time, Twitter. | (b) Cost, Twitter. | (c) Recourse, Twitter. |
| (d) Running time, covertype. | (e) Cost, covertype. | (f) Recourse, covertype. |
| (g) Running time, USCensus1990. | (h) Cost, USCensus1990. | (i) Recourse, USCensus1990. |

Figure 2: A comparison of the algorithms we consider in terms of running time (left column), cost of the solution (middle column), and recourse (right column).

set of facilities is therefore increasing (see Figure 1 in the supplementary material). The cost of MeyersonSingle generally has a linear dependency on the number of updates, though the slope is very gentle. This is also what our algorithm takes advantage off, broadly speaking by approximating the curve with a step function (adapted to handle insertions and deletions). The cost of our algorithm and MeyersonRec is basically indistinguishable, and in certain cases our algorithm fares even slightly better. The recourse of our algorithm is expectedly much better than MeyersonRec by a wide margin, and significantly worse than MeyersonSingle.

Finally, we ran our algorithm with multiple choices of facility cost $f$, and we observed that the recourse is almost independent of the both cost and running time of the algorithm, and only depends on the number of updates. This is consistent with tracking evolving data in time, where the underlying ground truth clustering also evolves in time.

**Acknowledgements.** Nikos Parotsidis is supported by Grant Number 16582, Basic Algorithms Research Copenhagen (BARC), from the VILLUM Foundation. Ce projet a bénéficié d'une aide de l'État gérée par l'Agence Nationale de la Recherche au titre du Programme FOCAL portant la référence suivante : ANR-18-CE40-0004-01.

## Footnotes

[1]$\widetilde{O}(\cdot)$ hides polylog factors.

[2]Under the common assumption that the ratio longest distance / shortest distance of the metric is polynomially bounded.

[3]We assume (as in [30]) that the minimum distance in the metric is 1 and the maximum $\Delta$ is bounded by a polynomial in $n^*$. Alternatively, our bounds can be stated with $\log \Delta$ instead of $\log n^*$.

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
