[Supplementary Material · FullyDynamicConsistentFacilityLocation-SupplementaryMaterial.pdf]

# Fully Dynamic Consistent Facility Location: Supplementary material

**Vincent Cohen-Addad**
CNRS & Sorbonne Université
vcohen@di.ens.fr

**Niklas Hjuler**
University of Copenhagen
hjuler@di.ku.dk

**Nikos Parotsidis**
University of Copenhagen
nipa@di.ku.dk

**David Saulpic**
Ecole normale supérieure
Sorbonne Univeristé
david.saulpic@lip6.fr

**Chris Schwiegelshohn**
Sapienza University of Rome
schwiegelshohn@diag.uniroma1.it

## 1 On Meyerson Algorithm

The Meyerson algorithm, presented in [4], is as follows.

---
**Algorithm 1** Meyerson($X, f$)

---
**Input:** A set of point $X$, an opening cost $f$
**Output:** A set of centers $S$ and an assignment $\phi$ of points to centers
 1: Let $S = \emptyset$
 2: **for all** point $x$ in $X$, in a random order **do**
 3:    Let $\delta = d(x, S)$
 4:    Add $x$ to $S$ with probability $\frac{\delta}{f}$
 5:    Define $\phi(x) = \underset{c \in S}{\arg\min}\, d(x, c)$
 6: **end for**

---

Some key properties of Algorithm 1 are summarized in the following lemma.

**Lemma 1.1** (from [4]). *An execution of Algorithm 1 has complexity $O(kn)$, where $k$ is the number of centers opened at the end. Moreover, the assignment given by $\phi$ has a cost $f \cdot |S| + \sum_{x \in X} d(x, \phi(x))$ that is at least $kf$, and in expectation is a 8-approximation of the optimal cost.*

## 2 Missing proofs from section 2

**Lemma 2.0** *Let $(X, d), (Y, d')$ be two metric spaces such that $X \subseteq Y$ and, restricted to $X$, $d = d'$. It holds that $C(X, OPT_Y) \leq 2C(X, OPT_X)$.*

*Proof.* Start from the optimal solution $OPT_Y$ on $Y$, with $k$ centers and where each point $x$ is assigned to a center $c_x$. Consider the solution on $X$ with $k$ centers, where the point closest to each

center of $\text{OPT}_Y$ is opened. Call $\psi(c)$ the center in $X$ opened instead of $c \in \text{OPT}_Y$.

$$2\text{C}(X, \text{OPT}_Y) \geq 2(kf + \sum_{x \in X} d(x, c_x))$$

$$\geq 2kf + \sum_{x \in X} d(x, c_x) + d(c_x, \psi(c_x))$$

$$\geq kf + \sum_{x \in X} d(x, \psi(c_x)) \geq \text{C}(X, \text{OPT}_X)$$

Where the last inequality holds because the set of centers $\psi(c)$ is a valid solution for Facility Location on the set $X$. $\qquad\square$

**Lemma 2.2** *Let $\text{OPT}_{before}$ be the optimal cost of an initial metric space $X$. After an arbitrary sequence of $n_i$ insertions and $n_d$ deletions of points in $X$, resulting in a metric space $X_{after}$, the optimal solution $\text{OPT}_{after}$ satisfies $C(X, \text{OPT}_{before})/2 - n_d \cdot f \leq C(X_{after}, \text{OPT}_{after}) \leq 2(C(X, \text{OPT}_{before}) + n_i \cdot f)$*

*Proof.* We remark that inserting a point can increase the optimal cost by at most $f$ (since opening a facility at the new point yields a solution of that cost).

Let $\text{OPT}'$ be the value of the optimal solution on $X \cup \mathcal{I}$ where $\mathcal{I}$ is the set of $n_i$ inserted points. By the previous observation, it holds that $\text{C}(X \cup \mathcal{I}, \text{OPT}') \leq \text{C}(X, \text{OPT}_{before}) + n_i \cdot f$. Let $X_{after}$ be the set of points after the sequence of $n_i$ insertions and $n_d$ deletions. Since deleting an arbitrary number of points can increase the optimal cost by at most a factor 2 (see Lemma 2.0), $\text{C}(X_{after}, \text{OPT}_{after}) \leq 2 \cdot \text{C}(X \cup \mathcal{I}, \text{OPT}') \leq 2(\text{C}(X, \text{OPT}_{before}) + n_i \cdot f)$.

Reversing the roles of $\text{OPT}_{before}$ and $\text{OPT}_{after}$ gives the other inequality: $\text{C}(X, \text{OPT}_{before}) \leq 2(\text{C}(X_{after}, \text{OPT}_{after}) + n_d \cdot f)$, and concludes the lemma. $\qquad\square$

**Proposition 2.4** *Any algorithm maintaining a constant-factor approximation for Facility Location requires $\Omega(n^*)$ update time and $\Omega(n)$ total recourse, where $n$ is the total number of updates.*

*Proof.* In the static case, $\Omega(n^2)$ time is required to find a constant factor approximation of Facility Location, even using randomization (see [5]). Hence, even in the incremental case, an amortized time $\Omega(n^*)$ is necessary.

For the recourse, for all $c$ we construct an instance that requires $\Omega(n)$ total recourse in order to maintain a $c$-approximation. Set $f = 1$, and let $u$ and $v$ be two vertices at distance $2c$. The stream of updates simply consist in adding $v$, then adding and removing $u$ $n$ times. Any $c$-approximation algorithm must add $u$ as a center at every time $t$ where $u \in X^t$: therefore the total recourse is $(n-1)/2 = \Omega(n)$. $\qquad\square$

**Lemma 2.5** *The algorithm from Section 2 can be adapted so that it maintains a $O(1)$-approximate solution, and each update takes time $O(n^* \cdot \log n^*)$ in the worst-case, with probability $1 - 1/n^*$.*

*Proof.* First condition on the event that every time we recompute from scratch we get a $\alpha$-approximation, with absolute value $\text{C}(X^{t_0}, S^{t_0})$. By Lemma 2.3, the algorithm from Section 2 maintains a $(8 \cdot \alpha + 4)$-approximate solution for the subsequent $\frac{\text{C}(X^{t_0}, S^{t_0})}{4 \cdot \alpha \cdot f}$ updates with probability $1 - \frac{1}{n^*}$. If the last solution was computed at time $t_0$, we begin to compute the next solution at time $t_0 + \frac{\text{C}(X^{t_0}, S^{t_0})}{4 \cdot \alpha \cdot f} - \frac{1}{8 \cdot \alpha} \frac{\text{C}(X^{t_0}, S^{t_0})}{4 \cdot \alpha \cdot f} = t_1$. This means that we have $\frac{1}{8 \cdot \alpha} \frac{\text{C}(X^{t_0}, S^{t_0})}{4 \cdot \alpha \cdot f} = x$ updates, before the new solution has to take over, at time $t_0 + \frac{\text{C}(X^{t_0}, S^{t_0})}{4 \cdot \alpha \cdot f}$. The algorithm starts to recompute a fresh solution during those $x$ updates, spending $O(n^* \log n^*)$ per update. We need to show two things: first, that this is indeed enough to recompute a solution, and second that two different recomputations do not overlap. In order to prove those two properties, it is necessary to ensure a deterministic bound on the complexity of Algorithm 1, and not only an expected one. For this, we first show a relation between $\text{C}(X^{t_1}, \text{OPT}^{t_1})$ and $\text{C}(X^{t_0}, \text{OPT}^{t_0})$.

$$\mathrm{C}(X^{t_1,t_1}) \leq 2 \cdot (\mathrm{C}(X^{t_0}, \mathrm{OPT}^{t_0}) + (t_1 - t_0) \cdot f)$$
$$\leq 2 \cdot (\mathrm{C}(X^{t_0}, \mathrm{OPT}^{t_0}) + \frac{\mathrm{C}(X^{t_0}, S^{t_0})}{4\alpha})$$
$$\leq 3 \cdot \mathrm{C}(X^{t_0}, \mathrm{OPT}^{t_0})$$

This relation shows that any execution of Algorithm 1 that open more than $128 \cdot \alpha^3 \cdot x$ centers is worthless. Indeed, opening more centers would yield a cost of at least $128 \cdot \alpha^3 \cdot x \cdot f = 4\alpha\mathrm{C}(X^{t_0}, S^{t_0}) \geq 4\alpha\mathrm{C}(X^{t_0}, \mathrm{OPT}^{t_0})$ – whereas the expected cost is at most $3\alpha \cdot \mathrm{C}(X^{t_0}, \mathrm{OPT}^{t_0})$.

Hence, among $\log 2n^*$ executions of Algorithm 1, one uses less than $128 \cdot \alpha^3 \cdot x$ centers with probability $1 - (1/n^*)^2$. The remark allows to stop the execution of all the ones that uses more centers, and the complexity is *deterministically* $\widetilde{O}(xn^*)$ for all these executions. Spread among $x$ updates, this is $\widetilde{O}(n^*)$.

We now prove that two recomputation do not overlap, i.e., that

$$t_0 + \frac{\mathrm{C}(X^{t_0}, S^{t_0})}{4 \cdot \alpha \cdot f} \leq t_1 + \frac{\mathrm{C}(X^{t_1}, S^{t_1})}{4 \cdot \alpha \cdot f} - \frac{1}{8 \cdot \alpha} \frac{\mathrm{C}(X^{t_1}, S^{t_1})}{4 \cdot \alpha \cdot f}.$$

This is equivalent to

$$\frac{1}{8 \cdot \alpha}\mathrm{C}(X^{t_0}, S^{t_0}) \leq (1 - \frac{1}{8 \cdot \alpha})\mathrm{C}(X^{t_1}, S^{t_1}).$$

Which is $\mathrm{C}(X^{t_0}, S^{t_0}) \leq (8 \cdot \alpha - 1)\mathrm{C}(X^{t_1}, S^{t_1})$. However, it holds that $\mathrm{C}(X^{t_0}, \mathrm{OPT}^{t_0}) \leq 3\mathrm{C}(X^{t_1}, \mathrm{OPT}^{t_1})$, hence $\mathrm{C}(X^{t_0}, S^{t_0}) \leq 3\alpha\mathrm{C}(X^{t_1}, S^{t_1})$, which concludes.

Notice that the analysis holds, conditioned to the fact that the last time the recomputation happened, the algorithms computed a $\alpha$-approximate solution. This happens with probability $1 - 1/n^*$ each time we recompute. Hence the time bound holds for each individual update with probability $1 - 1/n^*$. $\qquad\square$

## 3 Missing proofs from section 3

**Invariant 3.3** *The set $\mathcal{R}_i^t$ has size $O(k\log^2 n)$ and, with high probability, there exists $i$ such that $C_p(X^t, \mathcal{R}_i^t) = O(1) \cdot C_p(X^t, OPT^t)$.*

*Proof.* When $t$ is a multiple of $k$, this stems directly from Lemma 3.2. For sake of simplicity, let's assume that time 1 is the last time MeyersonCapped was called and that $t < k$.

$|\mathcal{R}^t|$ increased by $t$: it increases by 1 both in the case of point insertion and deletion. Therefore the size stays a $O(k\log^2 n)$.

Let $j = \lfloor \log \mathrm{C}_p(X^t, \mathrm{OPT}^t) \rfloor$: we prove that $\mathrm{C}_p(X^t, \mathcal{R}_j^t) = O(\mathrm{C}_p(X^t, \mathrm{OPT}^t))$. Let $f = \frac{L_j}{k(1+\log n)}$ be the facility cost for this instance.

The cost does not increase because of points additions, since the algorithm adds every new point directly to $\mathcal{R}^t$. We therefore ignore these newly added points in the following, and assume that only deletions occurred. In the following, the proof follows the line of the one in [1], taking into account the deleted points.

Let $c_1^*, ..., c_k^*$ be the optimal solution on $X^t$ and $C_i^*$ be the set of points assigned to $c_i^*$. Let $A_i^* = \sum_{x \in C_i^*} d(x, c_i^*)^p$ and $a_i^* = A_i^*/|C_i^*|$. For $j = 1, ..., \log n$ let $S_j$ be the set of points $x$ in $C_i^*$ such that $2^{j-1} \leq d(x, c_i^*) \leq 2^j$ together with the points $x \in X_1 \setminus X^t$ such that $c_i^*$ is their closest center (breaking ties arbitrarily). These points are exactly the one that have been deleted.

Consider the points in $S_j$, for $j \geq 1$. By linearity of expectation, the expected service cost before a point $c$ is opened is $f$. Any point $x$ arriving after a center is opened pays in expectation at most $3 \cdot 2^{p-1} d(x, c_i^*)^p$. However, it may happen that $c \in X_1 \setminus X^t$: in that case, the algorithm replaces it by $c'$, the closest point to $c$ in $X^t$. Hence any point $x$ arriving after $c$ pays at most $d(x, c')^p \leq 2^p d(x, c)^p \leq 6 \cdot 2^{2p-1} d(x, c_i^*)^p$, and the probability that a center is opened at $x$ is at most $3 \cdot 2^{2p-1} d(x, c_i^*)^p / f$.

Now consider the points in $S_1$. As before, the expected service cost paid before a point is opened is $f$. After a center is opened, the distance from any point $x$ to its nearest center is at most $d(x, c_i^*) + a_i^*$. In the case where the center is in $X_1$ but not in $X^t$, this cost becomes $2^{p-1}(d(x, c_i^*)^p + 2^p a_i^*)$. Hence the service cost is bounded by $2^{2p-1}(d(x, c_i^*)^p + a_i^*)$, and the probability to open a center at $x$ is at most $2^{2p-1}(d(x, c_i^*)^p + a_i^*)/f$.

The expected service cost for points of $C_i^*$ is therefore $f(1 + \log n) + 2^{2p-1} \sum_{x \in C_i^*} 3 d(x, c_i^*) + a_i^* \leq L/k + 2^{2p+2} C_p(C_i^*, \mathrm{OPT}^t)$. Summing over all clusters gives that the expected service cost is at most $L + 2^{2p+2} C_p(X^t, \mathrm{OPT}^t) \leq 2^{2p+3} C_p(X^t, \mathrm{OPT}^t)$. Moreover, the expected number of centers opened by the algorithm is $1 + \log n + 2^{2p-1}/f \sum_{x \in C_i^*} 3 d(x, c_i^*) + a_i^* \leq 1 + \log n + 2^{2p+2} A_i^*/f$. Summing again over all clusters gives that at most $k(1 + \log n)(1 + 2^{2p+2} \frac{C_p(X^t, \mathrm{OPT}^t)}{L}) \leq k(1 + \log n)(1 + 2^{2p+3})$ centers are opened.

Hence, with probability $1/2$, the service cost is $2^{2p+4} C_p(X^t, \mathrm{OPT}^t)$ and the number of centers is at most $2^{2p+4} k(1 + \log n)$. Since the algorithm opens exactly $2^{2p+4} k(1 + \log n)$ centers, this solution is found by the algorithm and $C_p(X^t, \mathcal{R}^t) \leq 2^{2p+3} C_p(X^t, \mathrm{OPT}^t)$. Since the algorithm maintains $O(\log n)$ independent execution of the algorithm, this cost is ensured with high probability, which concludes the lemma. $\square$

**Lemma 3.4** *Let* $OPT_{\mathcal{R}_i^t}$ *be the optimal solution in the weighted set* $\mathcal{R}_i^t$. *Then it holds that*
$$C_p(X^t, OPT_{\mathcal{R}_i^t}) \leq 2^{3p-1}(C_p(X, \mathcal{R}_i^t) + C_p(X^t, OPT^t))$$

This lemma is stated in [3], and generalize Theorem 2.3 in [2] to any value of $p$. For sake of completeness, we provide here a proof.

*Proof.* The proof stems from the two following inequalities:

- $C_p(X, \mathrm{OPT}_{\mathcal{R}_i^t}) \leq 2^{p-1}(C_p(X, \mathcal{R}^t) + C_p(\mathcal{R}^t, \mathrm{OPT}_{\mathcal{R}_i^t}))$
- $C_p(\mathcal{R}^t, \mathrm{OPT}_{\mathcal{R}_i^t}) \leq 2^{2p-1}(C_p(X, \mathcal{R}^t) + C_p(X^t, \mathrm{OPT}^t))$

Combining those two inequalities yields directly the lemma.

In order to prove those, we use the generalized triangle inequality $\forall x, y, z \in X, d(x, z)^p \leq 2^{p-1}(d(x, y)^p + d(y, z)^p)$. For any point $x$ and set $S$, we denote $S(x)$ the closest point in $S$ to $x$.

Let $x \in X$ and $y \in \mathcal{R}^t$ be its closest point in $\mathcal{R}^t$, such that $x$ contributes 1 in the weight of $y$.

We first prove the first inequality. It holds that $d(x, \mathrm{OPT}_{\mathcal{R}_i^t})^p \leq d(x, \mathrm{OPT}_{\mathcal{R}_i^t}(y))^p \leq 2^{p-1}(d(x, y)^p + d(y, \mathrm{OPT}_{\mathcal{R}_i^t}(y))^p)$. Since the point $y$ is weighted by the number of points assigned to it, summing over all $x, y$ gives exactly $C_p(X, \mathrm{OPT}_{\mathcal{R}_i^t}) \leq 2^{p-1}(C_p(X, \mathcal{R}^t) + C_p(\mathcal{R}^t, \mathrm{OPT}_{\mathcal{R}_i^t}))$.

For the second inequality, we consider the solution $S$ for the weighted set $\mathcal{R}^t$ consisting in the set of centers $\{\mathcal{R}^t(c), c \in \mathrm{OPT}^t\}$, and show that this solution has cost at most $2^{2p-1}(C_p(X, \mathcal{R}^t) + C_p(X^t, \mathrm{OPT}^t))$. Indeed, it holds that $d(y, S(y))^p \leq d(y, S(x))^p \leq 2^{p-1}(d(x, y)^p + d(x, S(x))^p)$. By definition of the set S, $d(x, S(x)) \leq 2 d(x, \mathrm{OPT}(x))$; therefore

$$d(y, S(y))^p \leq 2^{2p-1}(d(x, y)^p + d(x, \mathrm{OPT}(x))^p).$$

Summing over all $x, y$ proves the desired inequality, and concludes the proof. $\square$

**Theorem 3.1** *There exists a randomized algorithm that, given a metric space undergoing insertions and deletions of points, maintains a set of centers $S^t$ with $\widetilde{O}((n^* + k^2))$ update time such that, for any time $t$, $C_p(X^t, S^t) = O(1) \cdot C_p(X^t, OPT^t)$.*

*Proof.* Combining Invariant 3.3 and Lemma 3.4 proves the approximation ratio. We therefore turn on to the complexity bound.

Running the static algorithm on each coreset $\mathcal{R}_i^t$ takes time $\widetilde{O}(k^2)$, as the set $\mathcal{R}_i^t$ has size $O(k \log^2 n)$. Since the algorithm maintains $O(\log n)$ coresets of that size, the total cost for computations of the static algorithm is $\widetilde{O}(k^2)$.

The cost of maintaining $\mathcal{R}^t$ is similar to the one described in Section 2. The cost between two executions of MeyersonCapped is indeed $\widetilde{O}(nk)$: since $\widetilde{O}(k)$ centers are maintained, running Meyerson-Capped takes time $\widetilde{O}(nk)$. Moreover, a single execution of DeletePoint takes time $\widetilde{O}(n + (t_l^e - t_l^b) \cdot k$ (where $t_l^b$ and $t_l^e$ are the values of $t_l$ at that beginning and the end of the execution). Therefore, subsequent executions of DeletePoint take a total of $\widetilde{O}(nk)$ time. Since a computation of Meyerson-Capped occurs every $k$ updates, the amortized cost to maintain the sets $\mathcal{R}_1, ..., \mathcal{R}_{\log n}$ is $\widetilde{O}(n)$ per update, and the total amortized cost $\widetilde{O}(n + k^2)$. $\square$

## 4 Experiment: additional figures

Figure 1: Time for the whole Twitter dataset. MeyersonSingle is plotted in Orange, Our algorithm in blue and MeyersonRec is too slow to run on this dataset.