[Reviews · NeurIPS 2019]

Reviewer 1



The paper gives an algorithm for facility location that uses O(n log n) update time and has O(n) total recourse. It also gives an algorithm for k-means and k-median that uses \tilde{O}(n+k^2) update time, but the authors do not analyze the recourse for this algorithm; I believe the recourse is O(kn). The space used by all algorithms is O(n). By comparison, the naive algorithm for facility location uses O(n^2) update time and O(n^2) total recourse while the naive algorithm for clustering uses O(n polylog n) update time and O(kn) recourse, since a reclustering (such as by restarting an instance of the Meyerson online facility algorithm) can be computed upon every update when all the points are stored. The paper also performs empirical evaluations for their algorithm in comparison to Meyerson's online facility location algorithm. The facility location algorithm works by adding all new points as facilities for a consecutive number of steps, and then performing a maintenance step of finding a good subset of facilities among all points. When a point is deleted, the algorithm first considers whether the deleted point was a facility and if so, it opens a facility at the closest point. While the facility location algorithm is elegant, I'm not sure that the k-means and k-median algorithms obtain an advantage over the naive algorithms in space, update time, or recourse. I also think the discussion surrounding the empirical evaluations are insufficient or confusing. For example, I'm not sure why the blue line appears in the legend for Figures 2b, 2e, and 2h. Thus I think the paper can become a nice contribution but do not recommend acceptance in its current form. In particular, it is unclear to me whether it makes sense to run facility location algorithms on data sets considered: -- What does facility cost of 0.5 mean for Twitter/USCensus? -- There is no description of what the CoverType dataset actually is.

Reviewer 2



Originality: This work studies a problem variant related to variants studied in the past. It depends on some of these past results (e.g. Myerson’s algorithm). On the other hand, the problem seems a bit artificial; is there a practical use case for needing all three? For what problem is recourse necessary over the more standard Steiner tree problem? Quality: This work defends its algorithms with both theoretical proofs as well as empirical tests. In a sense, the experiments just confirm the obvious, since they’re comparing a tailored algorithm with the most obvious first attempts, but I can’t really fault the authors for doing the legwork. The work feels a bit incomplete in that it doesn’t really explain why the bounds on this algorithm are “right” for the problem. Clarity: I found the paper well-written. Significance: This is a nice starting point for future work to further develop by finding better algorithms or better lower bounds. Rebuttal: I have read the author's rebuttal, but I'm not sure the problem they study is completely aligned with the applications they suggest (clustering webpages, search queries, news articles, social networks, etc.). In these high scale applications, where super-efficient data structures become necessary, I would not expect the need for a data structure maintaining clusters to reflect the precise state of the world up to the last update, since there then there would be race conditions between updates and queries. In other words, it should be permissible to batch a bunch of updates. The results in this work appear optimized for the "a query appears after every update" ratio, and I wonder if better results are possible with batching. For example, is constant recourse per update necessary, or just constant recourse per batch? These questions could be answered by a better experimental section. Would this algorithm be good enough/what resources would it need to process one day's worth of tweets in a day? Also, the twitter data set seems to be the only data set even coming close to suggested applications.

Reviewer 3



The paper provides the first O(1)-competitive algorithm for facility location in adversarial arrival and departure model with reasonable recourse and update time. The algorithm works by showing that the optimum solution can not change dramatically: ignoring a static multiplicative factor of 2, one can show that deleting a client can decrease the cost by at most f, and adding a client can increase the cost by at most f. This way, one can divide the algorithm into phases: if at the beginning of a phase, the cost is Theta, then we can afford to run the algorithm for roughly Theta/f steps using trivial operations. Then at the end of each phase, we can recompute the solution from scratch. The algorithm for k-median/k-means is more complicated, since a single client can change the cost of the optimum solution dramatically. Thus, the algorithm can not achieve small recourse; the main focus on k-median/k-means is the update-time. The algorithm in the paper works by maintaining a small corset (for each guessed optimum value) that contains a good set of centers. The algorithm works in phases of k-steps. Inside each phase it only performs simple operations to the corsets. At the end of each phase, the corsets will be recomputed. The algorithm for the facility location problem is well written. However, I could not see the details of the algorithm and the analysis for the algorithm for k-median/k-means due to the writing style. In particular, I am confused with the procedure Delete point'' (see comments later); there is no intuition for this part at all. One reason is that the pieces of the algorithm are split in a somewhat arbitrary way into two parts: the main file and supplementary material. Moreover, there are some inconsistency of notations and typos. More details: there are many places in the supplementary material whether the authors used R_t, R^t_i and R_i. In the algorithm, should there only be R^t_i? In the Algorithm, DeltePoint, t_ell is an index for time and i is a time point between t_ell and |X| (indeed, this is a bad notation since i is used for something else). I am confused with the variable x_i. In the problem setting, x_i is the client that arrives or departs at time i. The total number of time steps so far is n, and the set of current clients is X. Then, what does the set {x_t, x_{t+1}, ...., x_|X|} represent? In the second column of figures for empirical analysis, the orange line is so think (is this a bug) that I can not see the blue line.

[Author Response · NeurIPS 2019]

We thank the reviewers for the thorough reviews.

**Reviewer #1 on the improvement over naive algorithms and Reviewer #2 on lower bounds:**   We would like to
point out that for k-median and k-means in general metric spaces, there is an $\Omega(nk)$ lower bound for computing
an $O(1)$-approximation that holds even for offline algorithms (see [2]). Thus, the naive algorithms achieve a total
running time of $\Omega(n(nk)) - \Omega(n^2 k)$ (by recomputing each time), whereas our algorithm achieves a total running time
of $O(n(n + k^2)polylogn)$   $O((n^2 + nk^2)polylogn)$ which is better for any $k = \omega(polylogn)$. For practical cases
where $k$ is often $n^c$ for some constant $c$, this is a significant improvement. Thus, our algorithm indeed has a better
running time than the naive algorithms. Notice that these bounds are tight up to $O(polylogn)$ factors of $k = O(\sqrt{n})$, as
inserting a new point in a general metric requires $O(n)$ time to describe the distances to other points. We will include
this discussion in the next version of our paper.

**Reviewer #1 and #3 on the experimental section:**   We acknowledge that the write-up of the experimental section is
not optimal, we realized that the blue line was hidden by the orange line, the new figures for the cost vs the number of
   updates for our algorithms vs MeyersonRec for USC, Twitter and Covtype are the following (in the preceeding order)

We will also add a description of the k-median/means algorithm within the first 8 pages.

**Reviewer #2 on problem justification:**   The study of online clustering dates back to the early 2000s. In many
practical scenarios, datasets that we would like to cluster are dynamic, for example webpages, search queries, news
articles, social networks, etc. Most of the literature has focused on the online model where a decision cannot be
undone or on the streaming model where there is a specific memory budget not to be exceeded. However, as observed
by Lattanzi and Vassilvitskii [1] the online model may appear too restrictive: if a bad decision has been made, it is
sometime fine to spend some time to correct it instead of suffering the bad decision (i.e.: keeping a bad clustering) for
the rest of the stream. However, spending too much time on the modification of the clustering may be counterproductive
and that's what we aim at capturing in this model: keeping a good clustering by spending the least among of time
and making as few changes to the current clustering as possible. For dynamic datasets, we do believe that the facility
location formulation of the problem is very well suited since the 'ground truth number of clusters' of the underlying
data may evolve and the facility location problem takes this into account through the facility cost.

**Reviewer #3 on k-means/k-median algorithm:**   In Algorithm DeletePoint, notice that $t_l$ is the latest point in time
that MeyersonCapped (of the particular copy of the algorithm) placed a center. Since time $t_l$, more points might have
been inserted for which we did not try to open a center as the number of centers would exceed the cap: this is the set
of points $x_t, x_{t+1}, \ldots, x_{|X|}$ from which we try to open a new center (to reach the cap again, after the deletion) and
recompute the assignment for the rest of the points.

Indeed, Algorithms MeyersonCapped and DeletePoint are executed on each of the $O(\log n)$ copies of the algorithm
identified by index $i$, and hence, index $i$ is not a good choice for iterating over the set of points $x_t, x_{t+1}, \ldots, x_{|X|}$.
We will change that in the revised version of our paper. Moreover, we are going to fix all inconsistencies in the
supplementary material.

# References

[1] S. Lattanzi and S. Vassilvitskii. Consistent k-clustering. In *ICML*, pages 1975–1984, 2017.

[2] R. R. Mettu and C. G. Plaxton. Optimal time bounds for approximate clustering. *Mach. Learn.*, 56(1-3):35–60,
June 2004.


[Meta-Review · NeurIPS 2019]

The authors give a set of provable algorithms for dynamic facility location and k-clustering under the additional assumption that the solution should remain relatively static from one iteration to the next. There is much to like about the paper, the theory is clean, and the experiments validate the theory and show that the constant factors are better than those predicted. At the same time, the paper lacks polish. As the reviewers pointed out, the empirical experiments were poorly written (fixed in the rebuttal), and the theory itself can be clarified (some of the algorithm blocks are missing and the text is hard to parse. I would encourage the authors to make a very careful revision of this work.